# Effects of a High-Fat Diet on Intestinal and Gonadal Metabolism in Female and Male Sea Cucumber *Apostichopus japonicus*

**DOI:** 10.3390/biology12020212

**Published:** 2023-01-29

**Authors:** Shuangyan Zhang, Xiaoshang Ru, Libin Zhang, David Gonçalves, Hongsheng Yang, Jialei Xu

**Affiliations:** 1CAS Key Laboratory of Marine Ecology and Environmental Sciences, Institute of Oceanology, Chinese Academy of Sciences, Qingdao 266071, China; 2Laboratory for Marine Ecology and Environmental Science, Qingdao National Laboratory for Marine Science and Technology, Qingdao 266237, China; 3CAS Engineering Laboratory for Marine Ranching, Institute of Oceanology, Chinese Academy of Sciences, Qingdao 266071, China; 4University of Chinese Academy of Sciences, Beijing 100049, China; 5Shandong Province Key Laboratory of Experimental Marine Biology, Qingdao 266071, China; 6Institute of Science and Environment, University of Saint Joseph, Macao SAR 999078, China; 7The Innovation of Seed Design, Chinese Academy of Sciences, Wuhan 430072, China; 8Zhongke Tonghe (Shandong) Ocean Technology Co., Ltd., Dongying 257200, China

**Keywords:** sex differences, high-fat diet, intestinal microbes, physiological metabolic responses, sea cucumber

## Abstract

**Simple Summary:**

The sea cucumber *Apostichopus japonicus* is an important aquatic invertebrate in the aquaculture industry. It has high nutritional and medicinal value and is popular with consumers, in particular in Asia. The high demand for the species has led to declines in wild populations and *A. japonicus* is listed as an endangered species by IUCN. Aquaculture of sea cucumber is important to safeguard the conservation of the species in the wild, and its successful reproduction in captivity is a prerequisite for industrial production. It is known that parental nutrient reserves, in particular of lipids, are directly related to the reproductive performance of sea cucumbers. In this study, we investigated the impact of a high-fat diet on the intestinal and gonadal metabolism of male and female *A. japonicus*. Diet lipid content significantly affected the intestinal microbiome and metabolite profile, and also interfered with gonadal metabolites. Dietary lipid supplementation enhanced the dominance of Proteobacteria in the intestines. Other microbes in the intestines responded to a high-fat diet, which may contribute to maintain intestinal homeostasis. The physiological and metabolic responses of sea cucumbers to a high-fat diet showed sex differences. Interestingly, a correlation between intestinal and gonadal tissue levels of polyunsaturated fatty acid was found. The results of this study indicate that the lipid content in diets can be differentially adjusted for male and female sea cucumbers to improve nutrition and promote reproduction. This data also contributes to a better understanding of the reproductive biology and sex differences of sea cucumbers.

**Abstract:**

Parental nutrient reserves are directly related to reproductive performance in sea cucumbers. This study focused on the lipid requirements of male and female sea cucumbers *Apostichopus japonicus* during the reproductive stage and analyzed their physiological responses to a high-fat diet (HFD). The intestinal lipid metabolites and microbiome profile changed significantly in animals fed with the HFD, as given by an upregulation of metabolites related to lipid metabolism and an increase in the predominance of Proteobacteria in the microbiome, respectively. The metabolic responses of male and female sea cucumbers to the HFD differed, which in turn could have triggered sex-related differences in the intestinal microbiome. These results suggest that the lipid content in diets can be differentially adjusted for male and female sea cucumbers to improve nutrition and promote reproduction. This data contributes to a better understanding of the reproductive biology and sex differences of sea cucumbers.

## 1. Introduction

Optimizing breeding technology and reproductive performance is crucial for the large-scale sustainable production of aquatic animals [1]. The nutrient reserves of parental animals have a significant role in determining their reproductive performance, with the quantity and quality of these reserves being associated with fecundity and gamete quality [2]. Thus, the energy source required for embryonic and larval development of most aquatic animals is dependent on the parental nutrient reserves [3]. Nutrient reserves are related to dietary intake and to the proportion of different nutrients in the diet, with essential dietary nutrients, such as lipids, directly affecting gonadal development in aquatic animals [4,5]. Lipids have complex functions in organisms, not only in energy storage but also in cell signaling and as membrane components [6]. Dietary lipids are an important source of nutrition, which can not only conserve dietary protein but also improve feed efficiency [7]. In addition, the amount of essential fatty acids significantly affects the reproductive performance of fish, while linolenic acid can affect the content of polyunsaturated fatty acids (PUFA) related to reproduction [8].

The growth and feeding efficiency of aquatic animals can be improved by appropriately increasing dietary lipids [9,10]. When a high-fat diet (HFD) contains a large amount of unsaturated fatty acids, the feed is easily oxidized during storage [11]. However, excessive intake of lipids can affect metabolic processes and significantly interfere with the intestinal microbial community structure [12,13]. In male mice, HFDs were found to alter the seminal fluid and gut microbiome [14]. Aquatic animals are also affected by HFDs, which might impair intestinal health by damaging the intestinal structure and the microbiota [15,16]. The microbiome is thought to influence metabolism and immune responses, which can affect both energy homeostasis and health [17]. Also, metabolic homeostasis was found to be affected by sex in most terrestrial animals, and sex differences in intestinal physiological metabolism might be an important factor leading to other sex-related differences in animals [18,19]. As an example, the physiological responses of zebrafish to HFDs, including changes in intestinal microbiota and fat deposition, were shown to be sex-specific [20,21,22]. Therefore, dietary lipids are important for both animal reproduction and intestinal health.

Sea cucumber *Apostichopus japonicus* is an important economic aquaculture product in China, with its production area increasing, and associated aquaculture technologies improving, over recent years [23]. The body wall of *A. japonicus* contains more proteins than lipids and is an excellent source of proteins, monounsaturated fatty acids, and n-3 polyunsaturated fatty acids [24]. The amount of eicosapentaenoic acid (EPA) in the body wall of juvenile sea cucumbers increases with dietary lipid levels [25], but the effects of the latter on reproduction have not been studied. Therefore, our research aimed to investigate the sex differences in lipid metabolism of sea cucumbers to determine whether males and females have different lipid nutritional requirements. For this, we manipulated the lipid content ingested by male and female sea cucumbers and analyzed the physiological changes in the intestines and gonads. Changes in intestinal microorganisms under the influence of an HFD were analyzed by 16s rDNA sequencing, and lipid metabolic pathways and related metabolites were studied using metabolomics approaches. Overall, the study aimed to investigate if the nutrition and reproduction of sea cucumbers could be improved by adjusting the fat content in the diet differently for males and females.

## 2. Materials and Methods

### 2.1. Experimental Design

Sea cucumbers *A. japonicus* were purchased from Zhongke Tonghe (Shandong) Ocean Technology Co., Ltd. (Dongying, China). Animals were allowed to acclimatize to stock ponds for 1 week before the start of the experiment, after which the wet weight of each individual was recorded. Sixty sea cucumbers were equally divided between two groups. The control group (C group) was fed with purchased compound feed mixed with three times the mass of sea mud. The HFD group (H group) was fed compound feed mixed with linseed oil (17:3 compound feed: linseed oil), and with three times the mass of sea mud. The ponds were continuously aerated, and the sea cucumbers were fed three times a day for 40 days. At the end of the experiment, all sea cucumbers were anesthetized with ice-cold water and their wet weights were recorded. They were then sacrificed, and their gonads, intestines, and other tissues removed and weighed. Subsequently, the gonads were checked under a microscope to determine the sex and reproductive stage, and samples from animals in the stage of gonadal proliferation were frozen in liquid nitrogen and stored at −80 °C for further analysis. Among them, 8 female and 8 male sea cucumbers were selected in H group and C group for subsequent experiments. The female sea cucumber intestines and gonads in the C group were marked as ICF and GCF, and in the H group were marked as IHF and GHF, respectively. The male sea cucumber groups were designated as ICM, GCM, IHM, and GHM, respectively.

### 2.2. Crude Fat and Fatty Acid Profile in Feed

#### 2.2.1. Detection of Crude Fat

To determine the composition of the feed used in the study, five replicate feed samples for each of the two groups were mixed, and the fat content of the mixed samples determined. Samples were weighed and placed in a tube to which 8 mL hydrochloric acid was added. The tubes were placed in a 70–80 °C water bath until the samples were completely digested. The tubes were then removed, and 10 mL ethanol was added and mixed. Once the tubes had cooled, the solution was transferred into a measuring cylinder with a plug, and 25 mL anhydrous ether was added, followed by oscillation. The solution was then left to stand for a few minutes and the supernatant was removed and put into a constant weight flask to which 5 mL anhydrous ether was added for repeated extraction. Anhydrous ether was recovered from the solution and the supernatant was dried in a water bath, followed by drying at 100 ± 5 °C to a constant weight. The crude fat content in the feed was calculated using Equation (1):(1)$$X=\frac{m_{2}-m_{1}}{m}\times 100$$
where, *m*_1_ is the original weight of the flask, *m*_2_ is the weight of the flask after final drying, and *m* is the weight of the tested feed sample.

#### 2.2.2. Analysis of Fatty Acid Profile

Feed samples from the two groups were weighed, placed in flasks, and successively mixed with 100 mg pyrogallic acid, several zeolites, 2 mL 95% ethanol, and 10 mL hydrochloric acid. Flasks were hydrolyzed for 40 min in a 70–80 °C water bath, and 10 mL 95% ethanol was added to flasks after they had cooled. Next, the hydrolysate was transferred to a separating funnel in which a 50 mL mixture of ether and petroleum ether had been added. The separating funnel was shaken for 5 min and left to stand for 10 min. The ether layer was collected into a 250 mL flask, dried in a water bath, and placed in an oven for 2 h at 100 ± 5 °C. Next, 2 mL 2% NaOH methanol solution was added to the flask for 30 min at 85 °C in a water bath, after which 3 mL 14% boron trifluoride methanol solution was added for another 30 min. Once the flask had cooled, 1 mL normal hexane was added; the flask was shaken for 2 min and left to stand for 1 h. After stratification, 100 μL supernatant was added to 1 mL normal hexane, and the solution was filtered through a 0.45 μm membrane. Finally, samples were analyzed by using a gas chromatograph mass spectrometer (Trace1310 ISQ, Thermo Fisher Scientific, Shanghai, China).

### 2.3. Intestinal Microbiota Analysis

#### 2.3.1. DNA Extraction, PCR Amplification of 16S rDNA Genes, and Sequencing

DNA was extracted from 32 samples of sea cucumber using a QIAamp DNA Mini Kit (Qiagen, Hilden, Germany). DNA purity was assessed on a 1% agarose gel and its concentration was measured using a NanoDrop 2100 device (Thermo Fisher Scientific). PCR amplification and sequencing of 16S rDNA was performed by Omicsmart (Guangdong, China). 16S rDNA was amplified by PCR using 341f341F(CCTACGGGNGGCWGCAG) and 806R(GGACTACHVGGGTATCTAAT) primers targeting the V4 region of the bacterial 16S rRNA gene. After purification of the PCR products, the sequencing library was constructed and sequenced using the Ion S5TMXL platform.

#### 2.3.2. Bacterial Community Composition, Alpha Diversity Analysis, and Function Prediction

The bacterial composition of the samples was analyzed with Uparse software, and the Effective Tags of all samples were clustered. The stacked bar chart of community composition was generated using the R project ggplot2 package (version 2.2.1). The Shannon Index was calculated in QIIME (version 1.9.1). Alpha index comparisons between groups were performed using Welch’s *t*-test. KEGG pathway analysis was performed using PICRUSt. Analysis of functional differences between groups was performed using the Welch *t*-test in R project Vegan package (version 2.5.3). Pearson correlation coefficient was used to analyze correlations between different microbiota and pathways.

### 2.4. Intestinal and Gonadal Metabolite Analysis

#### 2.4.1. Metabolite Extraction and LC-MS/MS Analysis

Tissue samples were added to tubes containing 1 mL extraction solvent and rotated for 30 s. The samples were then homogenized at 45 Hz for 4 min and treated for 5 min in an ice-water bath, with the procedure repeated three times. Next, the samples were incubated for 1 h at −20 °C and centrifuged for 15 min at 12,000 rpm and 4 °C. The supernatant was removed for UHPLC-QE Orbitrap/MS analysis. LC-MS/MS analysis was performed by Gene Denovo Biotechnology Co. (Guangzhou, China) using a UHPLC system (1290, Agilent Technologies, Santa Clara, CA, USA) with a UPLC HSS T3 column (2.1 mm × 100 mm, 1.8 μm) and Q Exactive Orbitrap MS (Thermo Fisher Scientific). The QE mass spectrometer registered MS/MS spectra on an information-dependent basis (IDA). Acquisition software (Xcalibur 4.0.27, Thermo) continuously evaluated full scan measures of the MS data as collected and recorded the acquisition of MS/MS spectra. MS raw data were converted to mzML format using ProteoWizard, and then processed using the R package XCMS (version 3.2) for retention time alignment, peak detection, and peak matching. After data processing using the MS/MS database, peak annotation was performed using OSI-SMMS (version 1.0, Dalian Chem Data Solution Information Technology Co., Ltd.).

#### 2.4.2. Differential Metabolite and Pathway Enrichment Analysis

Differential metabolites were analyzed by orthogonal partial least squares discriminant analysis. The threshold of variable importance in projection (VIP) was set to 1, and univariate analysis was performed in combination with the *p*-value of the *t*-test to analyze the differences in metabolites among the tissue samples. Metabolites with *p*-values < 0.05 and VIP ≥ 1 were considered to be differentially expressed between the experimental groups. The enrichment pathway analysis of differential metabolites was performed through KEGG databases, and metabolites related to lipid metabolism were screened by searching the KEGG B class. Pearson correlation coefficient was used to analyze correlations between differential metabolites and fatty acids in the tissue samples.

## 3. Results

### 3.1. Detection of Crude Fat and Fatty Acid Profile in Feeds

The proportion of crude fat content in the feed of the C group and H group was 2.6% and 6.5%, respectively (Table 1). Given the addition of linseed oil to the HFD, the C18:0 of the diet in the H group was more than five times that of the diet in the C group.

### 3.2. Analysis of Intestinal Microbiota in Sea Cucumbers

#### 3.2.1. Species Composition

The 16s rDNA sequencing results of the intestinal microbiota of the sea cucumbers showed that the top ten most-abundant phyla were Proteobacteria, Firmicutes, Bacteroidetes, Epsilonbacteraeota, Actinobacteria, Fusobacteria, Planctomycetes, Cyanobacteria, Patescibacteria, and Chloroflexi (Figure 1A). The top ten most abundant genera were *Pseudophaeobacter*, *Staphylococcus*, *Lutibacter*, *Arcobacter*, *Vibrio*, *Bacillus*, *Psychromonas*, *Salinicoccus*, *Propionigenium*, and *Oceanobacillus* (Figure 1A). Welch’s *t*-test results showed that the intestinal microbiome in the H group was significantly different from that of the C group, regardless of sex. Proteobacteria were significantly upregulated in the H group (Figure 1B,C), mainly as a result of the increased number of *Pseudophaeobacter* in this group. Regarding sex differences, they were not significant in the C group, whereas variation in Protobacteria underlined differences between males and females in the H group (Figure 1D). Among these, *Pseudophaeobacter* was more abundant in HF than in HM, whereas there were fewer *Methylobacterium* in HF than in HM.

#### 3.2.2. Alpha Diversity and Functional Prediction Analysis

Alpha diversity analysis can reflect the richness and evenness of microorganisms in a group. The Shannon index showed that species diversity was high in both the CM and CF groups, but there was no significant difference between the two groups (Figure 2A). However, there were significant differences in intestinal microbial diversity between the sexes in the H group. The microbial diversity in male and female sea cucumbers also decreased significantly after HFD treatment. PICRUSt, used for functional prediction, revealed 33 enriched pathways in four groups, mainly related to metabolism. The top ten pathways were carbohydrate metabolism, amino acid metabolism, metabolism of cofactors and vitamins, metabolism of other amino acids, metabolism of terpenoids and polyketides, xenobiotics biodegradation and metabolism, lipid metabolism, energy metabolism, replication and repair, and protein folding, sorting, and degradation (Figure 2B). The results of the Pearson correlation coefficient analysis showed that there was a significant negative correlation between the Proteobacteria and several pathways (Figure 2C), including transcription, signaling molecules and interaction, immune system, digestive system, and infectious diseases. In addition, *Pseudophaeobacter* was significantly negatively correlated with the pathways of replication and repair, protein folding, sorting and degradation, translation, transcription, signaling molecules and interaction, immune system, and infectious diseases (Figure 2C).

### 3.3. Analysis of Intestinal and Gonadal Metabolites in Sea Cucumbers

#### 3.3.1. Differences in Metabolites and Enrichment Pathways

The results of the differential analysis of *A. japonicus* intestinal and gonadal metabolites are shown in Figure 3A. The HFD had effects on the composition of intestinal and gonadal metabolites in sea cucumbers, with 45 upregulated and 100 downregulated metabolites in IHF versus ICF; 72 upregulated and 65 downregulated metabolites in IHM versus ICM; 23 upregulated and 66 downregulated metabolites in GHF versus GCF; and 36 upregulated and 44 downregulated metabolites in GHM versus GCM.

Sex also affected the intestinal and gonadal metabolites of *A. japonicus*, with 55 upregulated and 6 downregulated metabolites in ICF versus ICM; 13 upregulated and 12 downregulated metabolites in IHF versus IHM; 56 upregulated and 46 downregulated metabolites in GCF versus GCM; and 50 upregulated and 66 downregulated metabolites in GHF versus GHM. 

KEGG enrichment analysis of differential metabolites among each group showed that the top-20 enriched pathways were mainly related to metabolism (Figure 3B). Further analysis showed that most of these were lipid metabolism pathways, including fatty acid metabolism (KO00062, KO00071, KO01212), phospholipid metabolism (KO00564), and PUFA metabolism (KO00590, KO00591, KO00592). In addition, a few metabolites were enriched in organismal systems (KO04723) and human diseases (KO05231).

#### 3.3.2. Metabolites Related to Lipid Metabolism Pathways and Their Correlation with Diet

In fatty acid metabolism, changes were mainly seen in the activity of coenzymes (e.g., acetyl-CoA) involved in fatty acid metabolism, elongation, and decomposition pathways. Metabolites, such as choline, lecithin (PC), and lysophosphatidylcholine (LysoPC), varied significantly in each group, and were significantly enriched in the phospholipid metabolic pathway. PUFAs, such as α-linolenic acid, linoleic acid, arachidonic acid (ARA), and EPA, were also significantly altered (Figure 4A). Pearson correlation analysis (Figure 4B) showed that choline, cholesterol, Cer [d18:1/24:1(15Z)], 2-acetyl-1-alkyl-sn-glycero-3-phosphocholine, PE [18:3(9Z,12Z,15Z)/P-18:0], oleic acid, stearidonic acid, ARA, docosahexaenoic acid, cholesterol sulfate, PS (16:0/16:0), 11 types of PC, and 7 types of LysoPC were significantly correlated with dietary fatty acids. In addition, metabolites in gonads associated with dietary fatty acids included oleic acid, alpha-linolenic acid, cholesterol sulfate, and eight types of PC. Given the proportion of C14:0, C16:0, and C18:0 in the experimental diet, if these metabolites were positively correlated with C14:0 and C16:0, they were negatively correlated with C18:0, and vice versa.

## 4. Discussion

Dietary lipids impact the gonadal development of aquatic animals and the level of essential fatty acids in their diet is directly related to reproductive performance [4,8]. However, excessive intake of dietary lipids can significantly affect lipid metabolism and intestinal microorganisms [26]. The intestinal microbiota of aquatic animals is fluid and more sensitive to the influence of diet, compared with terrestrial animals, and lipid levels, lipid sources, and PUFA in dietary lipids can all affect the gut microbiota [27]. In the current study, the intestinal microbial community structure of the sea cucumber changed significantly after being fed an HFD. Notoriously, Proteobacteria, the most abundant group in all treatments, were significantly upregulated in both male and female sea cucumbers in the H group. These results are in agreement with previous studies in *A. japonicus* that showed that Proteobacteria is the main phylum in the intestines [28,29,30]. They also agree with studies in crustaceans where Proteobacteria in intestines were the dominant phylum and their abundance was positively correlated with dietary lipid content in Pacific white shrimp (*Litopenaeus vannamei*) and swimming crabs (*Portunus trituberculatus*) [31,32]. Taken together with the results of the current study, this suggests that the dominance of Proteobacteria in the intestines is common across different taxa and enhanced under a high lipid diet. Proteobacteria play an important role in the intestines by producing lipopolysaccharides that can trigger inflammation [33]. There was a significant negative correlation between the abundance of Proteobacteria and various pathways, including transcription, signaling molecules and interaction, immune system, digestive system, and infectious diseases, which might indicate that an HFD could affect intestinal functions in sea cucumber, such as digestive and immune function. Analysis at the genera level showed that the enrichment in Proteobacteria was mainly a consequence of a relative increase in *Pseudophaeobacter* under an HFD. In marine medaka (*Oryzias melastigma*), it was found that the stress response, glycerol metabolism, and metabolism of aromatic compounds were positively correlated with the abundance of *Pseudophaeobacter* [34]. It has also been reported that *Pseudophaeobacter* is rarely identified as a harmful microorganism and rather could serve as a potential antagonist of pathogenic microorganisms [35]. Accordingly, *Pseudophaeobacter* abundance was significantly negatively correlated with the pathways of replication and repair, protein folding, sorting, and degradation; translation, transcription, signaling molecules and interaction, immune system, and infectious diseases. This suggests that an increased abundance of *Pseudophaeobacter* in the intestines may have a protective effect against pathogens in sea cucumbers, although further studies would be needed to confirm this hypothesis. Some metabolites of host microbes can promote healthy intestinal function, and dietary components, such as lipids, can affect the intestinal microbiota and host fermentation products [36]. Thus, by changing the original intestinal microbiome structure, it is possible that an HFD contributes to maintaining intestinal homeostasis. Studies have also shown that the intestinal microbiome in mammals is sex-specific [37,38]. However, sex-based differences in the intestinal microbiome have rarely been reported in aquatic animals [39]. A comparison of the intestinal microbiota between male and female sea cucumbers showed no significant difference in community structure in control groups. However, when sea cucumbers were fed an HFD, the intestinal microbes of both sexes changed significantly but not similarly, such that the relative abundance of *Pseudophaeobacter* was higher in females than in males and the reverse was true for *Methylobacterium*. *Methylobacterium* can use single carbon substances, such as methanol and other methylated compounds, as substrates to provide carbon and energy via the serine cycle [40]. According to the changes observed for the intestinal microbiome, it can be confirmed that an HFD affects the intestinal physiology, that is, the colonization environment of intestinal microbes. It is possible that diet had a different impact in male and female physiological metabolism which, in turn changed intestinal microbes differently.

Dietary lipids have been shown to affect specific intestinal metabolites [36]. The current results corroborated these findings but also showed that an HFD affected gonadal metabolites. These metabolites were mainly involved in metabolic pathways, with a few related to disease pathways. Dietary lipid supplementation led to obvious changes in lipid metabolism, including fatty acid, phospholipid, and PUFA metabolism. To analyze the correlation of dietary lipids with intestinal and gonadal metabolites, metabolites related to lipid metabolism pathways were compared to the fatty acids in diets. The results showed that there were 29 and 11 metabolites related to lipid metabolism in the intestines and gonads respectively, that were significantly correlated with dietary fatty acids. In other words, these metabolites were clearly affected by dietary lipids. Moreover, the correlation between these metabolites and dietary lipids was related to the contents of C14:0, C16:0, and C18:0 in the diet. Choline is a vitamin that is a precursor of PC, which is a component of the cell membrane, and is often used as a feed additive because of this important physiological role [41,42]. Studies showed that dietary choline supplementation reduced fat deposition, whereas choline deficiency led to stunted growth, impaired lipid metabolism, and ruptured cell membranes [43]. When the lipid content in the diet was higher, the antioxidant capacity of aquatic animals decreased, which might be why an HFD reduced the choline content in the intestines and gonads of *A. japonicus* [44]. In addition, choline has an important role in ARA metabolism, and an HFD reduced not only the choline content, but also the ARA content in the intestines [45]. Another important PFUA, DHA, was also reduced in response to the HFD. According to the correlation analysis with dietary fatty acids, DHA and ARA were significantly negatively correlated with C18:0 in the diet. Although the content of C18:0 in the H group was approximately five times higher than that in the C group, there was a significant negative correlation between intestinal stearidonic acid and C18:0 in the diet. A positive correlation between stearidonic acid, ARA, and dietary DHA was found in blunt snout bream (*Megalobrama amblycephala*), while synergism between these metabolites was also found in the intestines of sea cucumber [46]. In the H group, the main addition was flaxseed oil, of which alpha-linolenic acid was the main component; unsurprisingly, the HFD increased alpha-linolenic acid levels in tissues in both males and females. A significant correlation between an HFD and alpha-linolenic acid was found in the gonads, which not only confirmed the reliability of dietary lipid effects in this experiment, but also suggests that a diet supplemented with linolenic acid affected gonadal metabolites. In rainbow trout studies, females were able to synthesize EPA and DHA from dietary linolenic acid [47]. In tilapia, increased dietary 18:3n-3 caused a corresponding increase in long-chain n-3 PUFAs in muscle lipids [48]. However, in the current study, only alpha-linolenic acid was increased, and DHA, EPA, and ARA in the intestines were significantly reduced. It was also reported that EPA in the body wall generally increased with dietary lipid levels in juvenile *A. japonicus* [25]. Therefore, it is possible that PUFAs are transferred from the intestines to other tissues in sea cucumbers [49]. Most metabolites were significantly influenced by diet, but sex also had an important role. In the current study, dietary lipids significantly affected DHA, EPA, and ARA in the intestines of sea cucumber, whereas sex had a significant impact on EPA and DHA in the gonads and EPA in the intestines. Meanwhile, DHA levels in the H group were found to be correlated in the gonads and intestines, which supports the previous hypothesis about the possible transfer of PUFA from intestines to gonads in sea cucumber [50]. In zebrafish, sex differences have been found not only in the intestinal microbiome, but also in fat deposition, pointing to sex differences in physiological metabolic responses to HFD [20,21,22]. In the intestines of sea cucumbers, an HFD affected the sphingolipid metabolic pathway and showed gender-based differences, whereas, in the gonads, triglyceride metabolism and biosynthesis of unsaturated fatty acid pathway were also impacted. Therefore, there were no sex differences in the intestines and gonads of the C group, whereas metabolites showed sex differences in the H group, suggesting that sea cucumbers show sex-based differences in physiological metabolic responses to an HFD.

## 5. Conclusions

This study aimed to determine whether male and female sea cucumbers *A. japonicus* have different nutritional requirements for lipids by exploring sex differences in lipid metabolism. It was found that an HFD significantly affected the intestinal microbiome and metabolite profile, and also interfered with gonadal metabolites. Dietary lipid supplementation enhanced the dominance of Proteobacteria in the intestines. Other microbes showed changes in the intestines, which may contribute to maintaining intestinal homeostasis. Physiological and metabolic responses of sea cucumbers to an HFD showed sex differences, which has probably led to differences in the intestinal microbiome of male and female sea cucumbers. However, it is also possible that the changes in physiological metabolism are caused by the direct influence of diet on intestinal microbes. Interestingly, an intestinal-gonadal tissue correlation was found in PUFA levels. These results are of relevance to the reproductive biology of sea cucumbers as they show that an HFD impacts metabolism, the intestinal microbiome and reproductive physiology. The study also demonstrates that an HFD has different impacts in the physiology of male and female sea cucumbers. Overall, the data may contribute to improving the nutrition and reproductive performance of sea cucumbers, highlighting the need to account for sex differences when considering the introduction of dietary changes.

## Figures and Tables

**Figure 1 biology-12-00212-f001:**
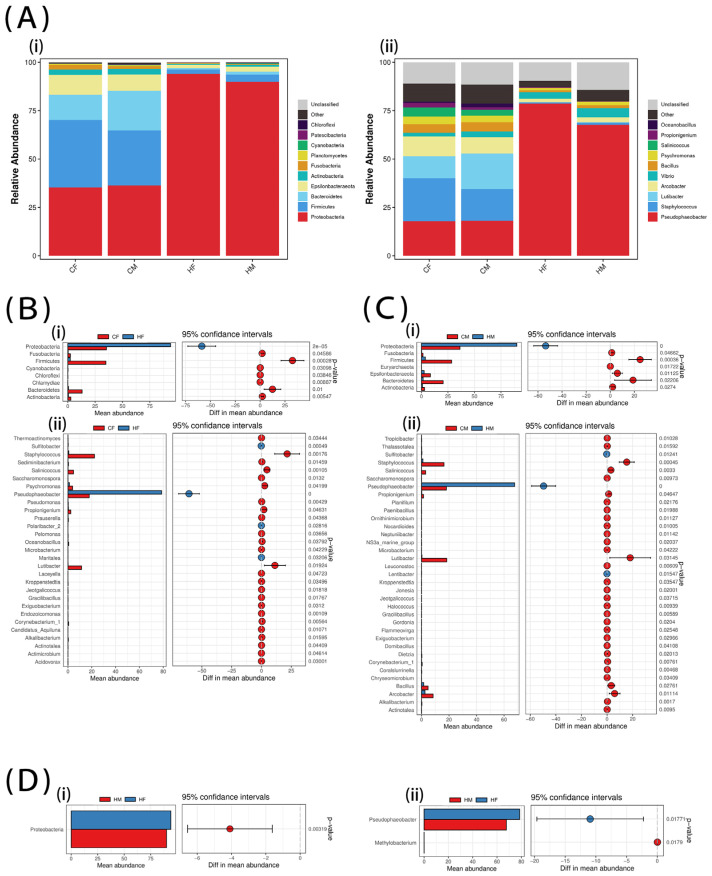
Species composition analysis of intestinal microbiota in sea cucumbers [(i): phyla; (ii): genus]. (**A**) The top ten most-abundant microbiota in each experimental group; (**B**) Microbiota that were significantly different between CF and HF. (**C**) Microbiota that were significantly different between CM and HM. (**D**) Microbiota that were significantly different between HM and HF.

**Figure 2 biology-12-00212-f002:**
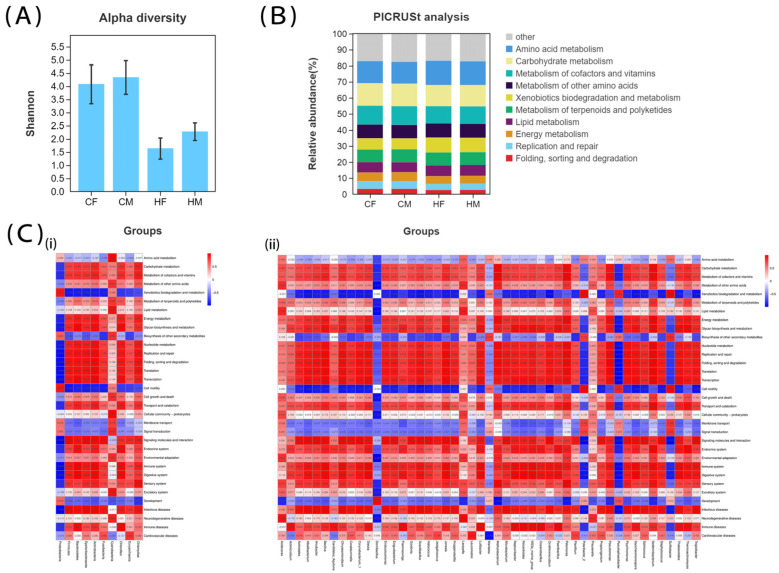
Alpha diversity and function analysis of intestinal microbiota in sea cucumbers. (**A**) Alpha diversity by Shannon index. (**B**) The top ten most enriched microbiota pathways. (**C**) Functional correlation analysis of microbiota that were significantly different between experimental groups [(i): phyla; (ii): genus].

**Figure 3 biology-12-00212-f003:**
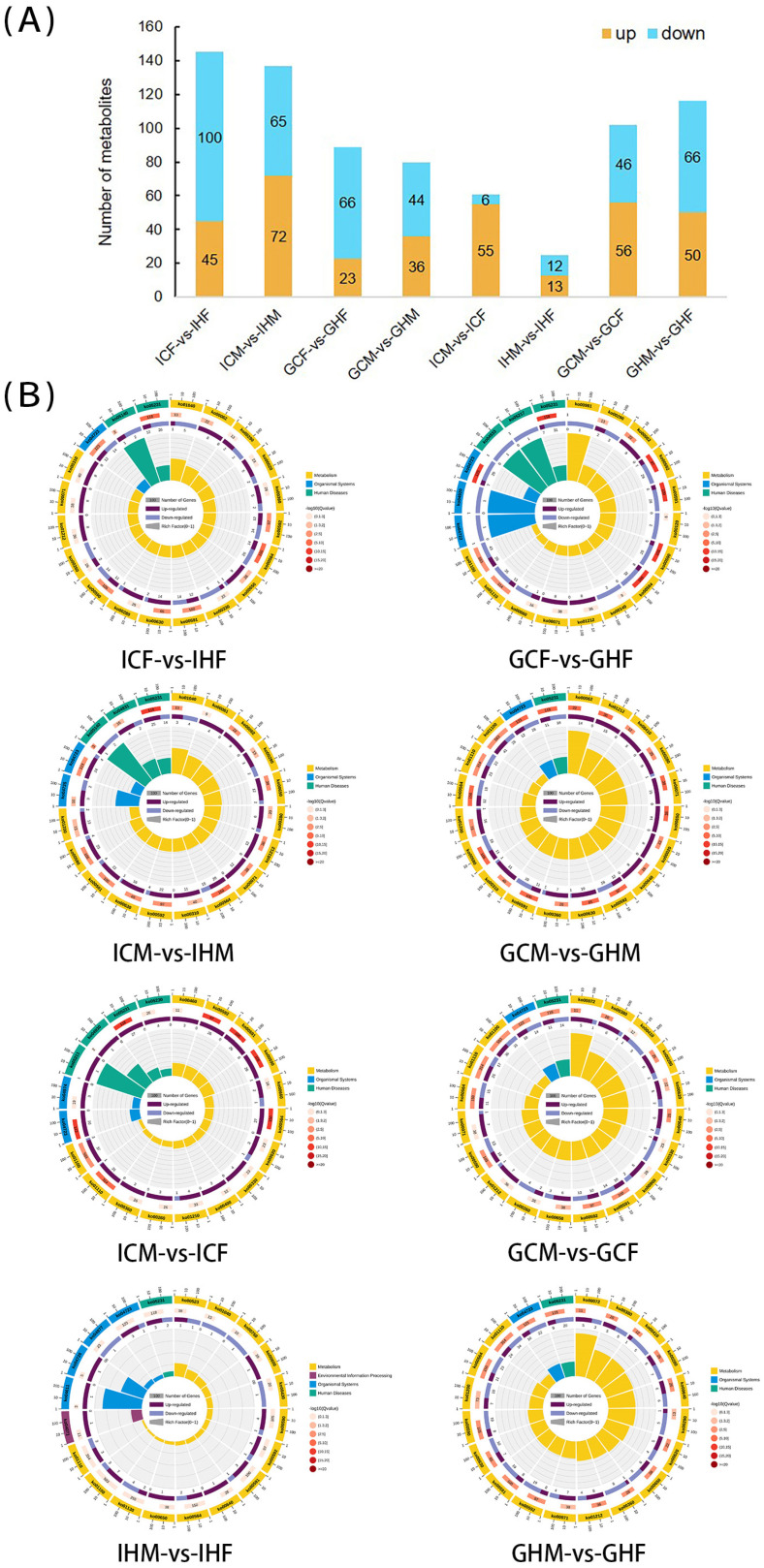
Differential intestinal and gonadal metabolites and enrichment pathways in sea cucumbers. (**A**) Comparison of differential metabolites between experimental groups. (**B**) The top-20 most-enriched differential metabolite pathways between the experimental groups.

**Figure 4 biology-12-00212-f004:**
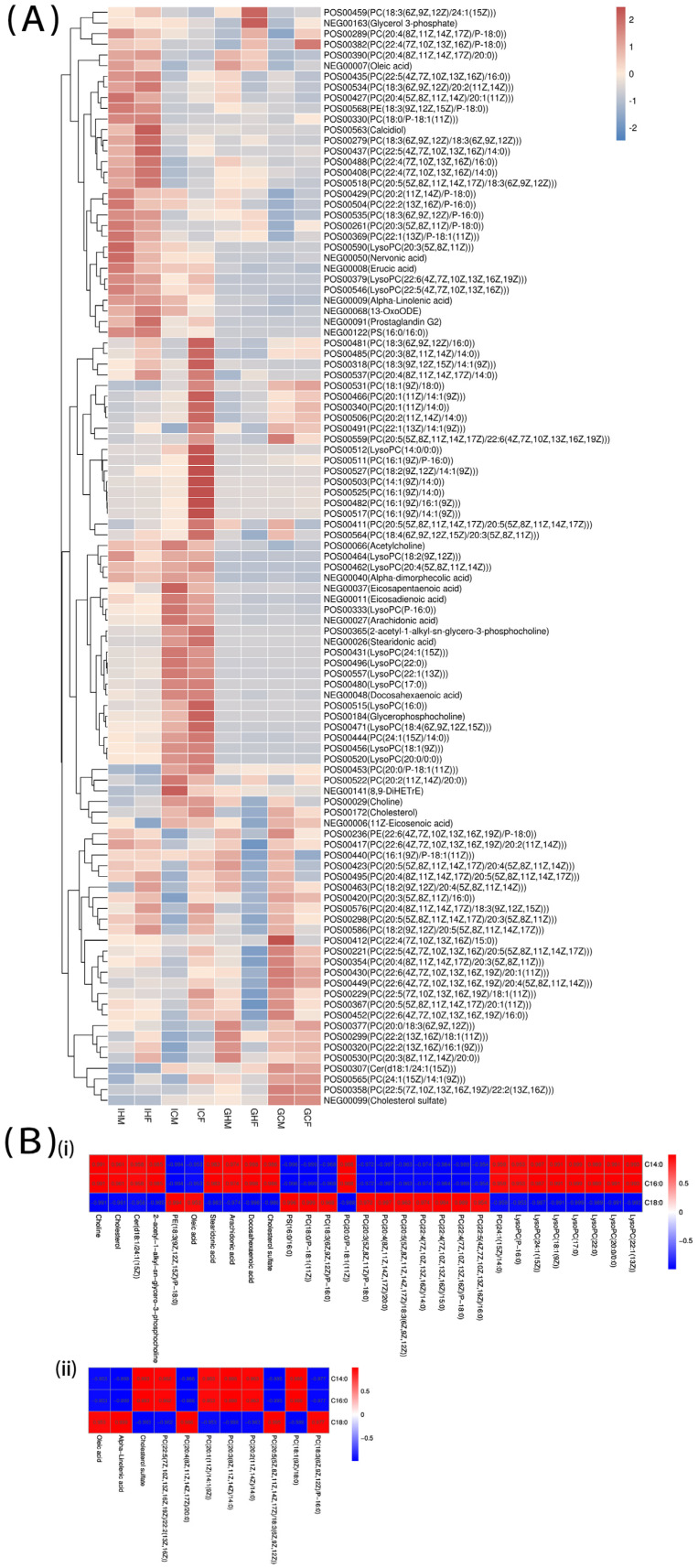
Intestinal and gonadal metabolites related to lipid metabolism pathways in sea cucumbers and their correlation with fatty acids in experimental diets. (**A**) Heatmap of metabolites related to lipid metabolism pathways. (**B**) Lipid metabolites significantly associated with fatty acids in experimental diets [(i): intestines; (ii): gonads].

**Table 1 biology-12-00212-t001:** Levels of crude fat and fatty acids in experimental diets.

Content (g/100 g)	Experimental Group	
	C	H
Crude fat	2.6	6.5
C14:0	0.0095	N/D
C16:0	0.0942	0.0816
C18:0	0.0103	0.0583

N/D (not detected): fatty acids were not detected when the content was <0.0033 (g/100 g).

## Data Availability

Data is contained within the article or Appendix A.

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
