# Peer review of "Effects of a High-Fat Diet on Intestinal and Gonadal Metabolism in Female and Male Sea Cucumber Apostichopus japonicus"

_biology, 2023, doi:10.3390/biology12020212_

Round 1

Reviewer 1 Report

This manuscript shows detailed datasets and shifts in microbiome and metabolomics after treatment of HFD between female and male sea cucumbers. All these data are convincing and publishable. The major concern is that it is not clear what is the scientific question the authors want to address. Several specific questions need to be solved before publishing:

1.      How would be the shifted metabolites affecting the human beings after intaking of the sea cucumbers?

2.      With your findings in this study, what do you want to disseminate/highlight?  For the HFD with linseed oil, it’s confused that the sea cucumbers were treated with such high artificial fat, which is abnormal in nature.

Reviewer 2 Report

The manuscript presents quite interesting data on the effect of the addition of linseed oil to the diet, and examined males and females after 40 days for their microbiomes and metabolites. That part of the study was well done and discussed. However, they claim that their results are significant to understand/improving reproduction and nutrition, yet they don't even show at what reproductive stage the animals were! That is something that could strongly impact the results, yet despite looking at the gonad to determine sex, surely they could, and should state the stage of development. If the two groups had different development, that needs to be considered in the results. They talk about the impact to physiology at the conclusions, but what physiological measures were undertaken? To have this paper suitable for publication, these need to be addressed. 

Round 2

Reviewer 1 Report

Thanks for the authors' serious consideration/response for the comments and this is the 2nd time to review this manuscript. I do think the current version could meet the publication criteria. 

Reviewer 2 Report

I have agreed to review this paper as I was interested in the physiological effects, and by removing physiology from the title, you now cover what you did, which is fine but less interesting.